# Characterization of Two Mouse *Chd7* Heterozygous Loss-of-Function Models Shows Dysgenesis of the Corpus Callosum and Previously Unreported Features of CHARGE Syndrome

**DOI:** 10.3390/ijms231911509

**Published:** 2022-09-29

**Authors:** Stephan C. Collins, Valerie E. Vancollie, Anna Mikhaleva, Christel Wagner, Rebecca Balz, Christopher J. Lelliott, Binnaz Yalcin

**Affiliations:** 1Inserm UMR1231, University of Burgundy Franche-Comté, 15 Boulevard Maréchal de Lattre de Tassigny, 21070 Dijon, France; 2Wellcome Sanger Institute, Hinxton CB10 1SA, UK; 3Center for Integrative Genomics, University of Lausanne, 1015 Lausanne, Switzerland; 4Institute of Genetics and Molecular and Cellular Biology, UMR7104, 67400 Illkirch, France

**Keywords:** neurodevelopmental disorders, CHARGE syndrome, dysgenesis of the corpus callosum, mouse models, CHD7

## Abstract

CHARGE syndrome is a rare congenital disorder frequently caused by mutations in the chromodomain helicase DNA-binding protein-7 *CHD7*. Here, we developed and systematically characterized two genetic mouse models with identical, heterozygous loss-of-function mutation of the *Chd7* gene engineered on inbred and outbred genetic backgrounds. We found that both models showed consistent phenotypes with the core clinical manifestations seen in CHARGE syndrome, but the phenotypes in the inbred *Chd7* model were more severe, sometimes having reduced penetrance and included dysgenesis of the corpus callosum, hypoplasia of the hippocampus, abnormal retrosplenial granular cortex, ventriculomegaly, hyperactivity, growth delays, impaired grip strength and repetitive behaviors. Interestingly, we also identified previously unreported features including reduced levels of basal insulin and reduced blood lipids. We suggest that the phenotypic variation reported in individuals diagnosed with CHARGE syndrome is likely due to the genetic background and modifiers. Finally, our study provides a valuable resource, making it possible for mouse biologists interested in *Chd7* to make informed choices on which mouse model they should use to study phenotypes of interest and investigate in more depth the underlying cellular and molecular mechanisms.

## 1. Introduction

CHARGE syndrome (MIM #214800) is a rare congenital disorder (present from birth in about 1 in 10,000 newborns) affecting many areas of the body and leading to complex malformations encompassing (but not limited to) coloboma, heart defects, atresia of the choanae, restricted growth and development, genital hypoplasia and ear anomalies, and deafness, hence the acronym CHARGE. The majority of cases are linked to heterozygous loss-of-function (LoF) mutations in the chromodomain helicase DNA binding protein coding *CHD7* gene [1]. Although brain-related abnormalities are not used as landmarks for clinical diagnosis of CHARGE syndrome, patients show neurodevelopmental disorders such as autism spectrum and hyperactivity disorders. These are associated with brain structural anomalies including hindbrain malformations such as abnormal cerebellar vermis and abnormal cerebellar foliation, malformations in the hippocampus, the hypothalamus and the posterior fossa, cortical atrophy (in particular frontal lobe hypoplasia), ventricular enlargement and dysgenesis of the corpus callosum [2,3,4,5]. Yet, the clinical presentation of individuals with CHARGE syndrome remains heterogeneous, with some baring atypical phenotypes, which has pushed several times for the refinement of the full phenotypic spectrum [2,6,7].

To explain why some individuals present variable features, CHARGE syndrome has been modeled in a large variety of organisms, ranging from (*C. elegans*, *D. melanogaster*, *X. laevis*, *D. rerio* to *M. musculus* (the house mouse), whilst the function of CHD7 has been investigated down to unicellular organisms such as yeast. Several mouse models have been established to better understand the role of CHD7 in CHARGE syndrome, especially given the wide range of congenital abnormalities that can occur. Gene-trapped ES cell lines, N-ethyl-N-nitrosourea (ENU) mutagenesis, targeted *Chd7* gene mutations, and multiple floxed alleles of the gene are among the mouse models summarized in a comprehensive review [7].

The *Chd7^tm2a(EUCOMM)Wtsi^* mouse, hereafter referred as *Chd7^+/tm2a^* for heterozygous or *Chd7^tm2a/tm2a^* for homozygous knock-out (KO) animals, was generated via the International Mouse Phenotyping Consortium (IMPC) using the KO-first allele strategy and consists of a reporter-tagged insertion with conditional potential that results in the excision of a critical exon (here exon 3, Figure 1A) predicted to stop transcription/translation after exon 2 [8]. The sub-viability of these mice necessitated the use of a mixed genetic background (C57BL/6N; 129S5), whilst the ENU mutant strain *Chd7^+/Whi^* was bred on a pure C3HeB/FeJ background. The *Chd7^tm2a^* constitutive model, and corresponding conditional line *Chd7^tm2d(EUCOMM)Wtsi^* which can be derived from it, has been extensively used to investigate CHD7-dependent cardiogenesis [9,10,11], neuronal function [12,13,14,15,16,17,18] and deafness [19,20].

The *Whirligig* mouse, referred as *Chd7^+/Whi^* for heterozygous or *Chd7^Whi/Whi^* for homozygous mutant animals, is one of eleven mouse models generated by a large-scale ENU-induced mutagenesis program [21]. Both the *Chd7^+/tm2a^* and *Chd7^+/Whi^* mutants are LoF models. Whilst *Chd7^+/tm2a^* is a constitutive KO with tissue or cell-specific conditional potential, the ENU-induced *Chd7^+/Whi^* mutant contains a point mutation at exon 11 (out of 38), causing a premature stop codon at p.Trp973*, which is relatively early in a protein that counts 2986 amino acids. Hence, this mutation is likely to lead to nonsense-mediated mRNA decay. The original phenotypic screen investigating *Chd7^+/Whi^* mutants identified deafness due to semicircular canal defects which were fully penetrant, and occasional heart defects, choanal atresia, cleft palate and eye defects [21]. Given the high embryonic expression of *Chd7* in the olfactory bulb and hypothalamus, much of the neuroanatomical characterization has focused on the circuitry of smell and reproduction [22]. More recently, the *Chd7^+/Whi^* model was used to demonstrate the involvement of *Chd7* in regulating genes involved in neural crest and axon guidance [23], and proliferation of neural stem cells [24].

In this study, we address the question of why individuals diagnosed with CHARGE syndrome show phenotypic variability by exploring and comparing *Chd7* function in two adult mouse models (*Chd7^+/Whi^* and *Chd7^+/tm2a^*) engineered on different genetic backgrounds to mimic the human condition. Using ultra-standardized pipelines for the characterization of brain neuroanatomy [17,25] and whole-body phenotypes including body weight, body composition, blood chemistry, neurological and behavioral assessments, we carried out a systematic evaluation of both mouse models and found that, overall, *Chd7^+/Whi^* mice engineered on a pure genetic background were more severely affected than *Chd7^+/tm2a^* mixed-background mice. Interestingly, while both models exhibited defects in line with patient data, we identified previously unreported features for CHARGE syndrome, including reduced levels of insulin, which could potentially be tested as a new biomarker for CHARGE syndrome.

## 2. Results

Since no previous mouse studies have reported the implication of *Chd7* in the context of brain neuroanatomy in adulthood, we used a highly robust approach for the neuroanatomical assessment and analysis of 41 brain parameters distributed across 14 developmentally distinct brain regions. We focused on five main brain categories: brain size, commissures (callosal, mammillothalamic tract, internal capsule, optic tract, fimbria), ventricles (lateral and third dorsal), cortex (primary motor, secondary somatosensory, retrosplenial granular cortex), and subcortex (hippocampus, amygdala, habenular). Both the *Chd7^Whi^* and the *Chd7^tm2a^* heterozygous mouse colonies presented major anomalies when compared to littermate controls, in particular, absence of midline crossing leading to dysgenesis of the corpus callosum (Figure 1C–E). Interestingly, only the *Chd7^+/tm2a^* model displayed a completely penetrant phenotype with three out of three mutants exhibiting corpus callosum dysgenesis (Figure 1D), whilst the *Chd7^+/Whi^* model revealed two normal and one affected brain (the latter is shown in Figure 1E). The retrosplenial granular cortex and the hippocampus were abnormally shaped, with their size affected most likely due to the severe midline crossing defects (Figure 1D,E). The size of the third dorsal ventricle (D3V) was severely enlarged in the *Chd7^Whi^* model (indicated by a red asterisk in Figure 1E). In *Chd7^+/tm2a^* mice, we were also able to quantify an increase of 100%, on average, in the size of the mammillothalamic tract on both sides in *Chd7^+/tm2a^* mice (*p* < 0.002), a decrease of 32% in the size of the internal capsule (*p* < 0.01), and an increase of 41% in the size of the habenular nucleus (*p* < 0.01) (Figure 1C).

Extensive, whole organism phenotyping in new cohorts identified several common features between the two mutant lines, which are summarized in Figure 2.

At P14 age, mouse survival was assessed from successfully genotyped mice originating from several litters and derived from a “heterozygous-by-heterozygous” breeding scheme (Figure 1B). In *Chd7^tm2a^* mice, we obtained no homozygous mice suggesting that *Chd7* is essential for viability. To determine the window of death, we carried out a recessive lethality screen at mouse embryonic day 14.5 (E14.5). We looked at 36 embryos, 2 of which were homozygous, including 1 homozygous that displayed oedema and craniofacial defects, suggesting that homozygous mice died before the mid-gestational stage. For *Chd7^Whi^*, the results were similar in terms of increased lethality but published elsewhere [21] and adapted for comparison purposes in Figure 3A. Craniofacial defects as well as other dysmorphological features were absent in both *Chd7^+/Whi^* and the *Chd7^+/tm2a^* mice (Figure 2 and Appendix A). Both *Chd7^+/Whi^* and the *Chd7^+/tm2a^* colonies displayed lower body weight from early life to adulthood (Figure 3B), and exhibited shorter length at 14 weeks of age (Figure 3B insert). Body composition analysis showed reduced lean mass by 16% in both mutant lines (*p* = 0.04 and *p* = 0.005 in *Chd7^+/Whi^* and *Chd7^+/tm2a^* mice, respectively) and reduced fat mass (−30%, *p* = 0.04) and fat percentage (−28%, *p* = 0.03) specifically in *Chd7^+/Whi^* mice. Bone mineral content was marginally reduced in *Chd7^+/tm2a^* mice (−15%, *p* = 0.05) (Appendix A). *Chd7^Whi^* mice were particularly active as reflected by increased locomotor activity evaluated by the number of square crosses in the modified SHIRPA (Figure 2). Mice also displayed head bobbing (repetitive up and down movement of the head) and trunk curling. The *Chd7^+/tm2a^* phenotype was milder with no excessive activity scored and normal locomotor activity during the modified SHIRPA protocol and in the open-field paradigm (Figure 3C). Hyperactivity was confirmed in *Chd7^+/Whi^* mutants both at the periphery and at the center of the open-field apparatus (Figure 3C). Grip strength unadjusted for body weight was lower in both lines, but more pronounced in *Chd7^+/Whi^* mutants (Figure 3D). Sensitivity to heat was normal in *Chd7^+/tm2a^* mutants but lower in *Chd7^+/Whi^* as evidenced by the hot plate test (+47% in response latency, *p* = 0.02) (Appendix A).

Next, we assessed metabolism and found that *Chd7^+/Whi^* mutants displayed elevated VCO_2_ and VO_2_ production and consumption, respectively, resulting in a normal respiratory exchange ratio (Appendix A). Based on average values of glucose tolerance tests, glucose homeostasis was normal in both *Chd7^+/Whi^* and *Chd7^+/tm2a^* mutants. However, the *Chd7^+/Whi^* line displayed a dichotomic response with glycaemia either above or below corresponding controls (Figure 4A–C). Interestingly, *Chd7^+/Whi^* and *Chd7^+/tm2a^* mutants showed low fasted insulin levels (Figure 4D), although only the *Chd7^+/Whi^* line reached significance (*p* = 0.001). Standard blood chemistry revealed stronger effects in the *Chd7^+/Whi^* mutant mice also with a reduction in potassium, triglycerides, cholesterol, high-density lipoprotein (HDL), low-density lipoprotein (LDL), non-esterified free fatty acids (NEFA), glycerol, aspartate aminotransferase, albumin, calcium and magnesium, but increased urea levels. *Chd7^+/tm2a^* mutants also displayed lower LDL and total protein levels but uniquely featured lower alanine aminotransferase whilst alkaline phosphatase and uric acid levels were higher than matched controls (Figure 4E and Appendix A).

X-ray imaging did not reveal skeletal malformations (Appendix A). Stress Induced hyperthermia showed that *Chd7^+/Whi^* mutants had a lesser response (+1.42 °C ± 0.13 versus +1.84 °C ± 0.16 in mutants and controls, respectively) although reaching marginal significance (*p* = 0.07). The same trend was seen in *Chd7^+/tm2a^* mutants with +1.61 °C ± 0.27 versus +2.1 °C ± 0.18 in mutants and controls, respectively (*p* = 0.06).

Eye morphology had a low penetrance hit, with 5 out of 14 *Chd7^+/tm2a^* mutants found to have an abnormal pupil position or shape in the left or predominantly right eye. *Chd7^+/tm2a^* mutants also had opaque lenses, although not in the same animals scored with pupil abnormalities. None of the 14 controls used in this assay were positive for any of these eye phenotypes. Interestingly, the *Whirligig* mutant did not exhibit eye phenotypes in opposition to a previous report [21], although it should be emphasized their study showed incomplete penetrance.

Haematology data highlighted reduced haematocrit as a common feature for both mutant lines. In the *Chd7^+/tm2a^* mutant line exclusively, white blood cell, heamoglobin and red blood cell distribution width were also lower compared to controls (Appendix A). Peripheral blood leukocytes were measured and expressed as a percentage for each cell type. There was an increase in natural killer (NK) cells in both mutant lines (2.1 ± 3.2 versus 3.5 ± 0.5 in controls and *Chd7^+/tm2a^* and 3 ± 2.8 versus 4.1 ± 0.8 in controls and *Chd7^+/Whi^*, respectively) although only reaching statistical significance in the *Chd7^+/tm2a^* mutant line (Figure 4F). In contrast, the percentage of monocytes were lower in both lines (7.7 ± 8.1 versus 5.8 ± 1.4 in WT and *Chd7^+/tm2a^* and 8.7 ± 8.1 versus 3.7 ± 0.6 in WT and *Chd7^+/Whi^*, respectively) with significance levels reached only in *Chd7^+/Whi^* animals (Figure 4G).

## 3. Discussion

We report, here, an extensive characterization of two mouse models of CHARGE syndrome, namely *Chd7^+/tm2a^* and *Chd7^+/Whi^*, encompassing metabolic, behavioural, immunological and morphological parameters.

Our findings are important for three main reasons. First, we report major neuroanatomical phenotypes using a precise and quantitative assessment of brain structural defects in two heterozygous LoF mouse models of CHARGE syndrome. Aside from studies reporting sites of normal *Chd7* expression in the murine brain [21,22], only two studies have previously reported structural neuroanatomical findings. A conditional mouse study of nestin-Cre *Chd7^tm2a^* reported defects pertaining to the cerebellum with cerebellar hypoplasia, by far the strongest phenotype [14]. Another study of a mouse identified in an ENU mutagenesis screen described defects in the developing murine brain using a qualitative description of dilated lateral and third ventricles, with hippocampus atrophy and corpus callosum crossing failure [26]. Consistently, our findings agree with the notion that *Chd7* plays a major role in midline crossing by contralateral cortical neurons projections along the corpus callosum. Interestingly, both *Chd7^+/tm2a^* and *Chd7^+/Whi^* mutants display not only lack of midline crossing, but also an abnormally shaped and enlarged retrosplenial granular cortex which seems to invaginate along the sagittal axis and expand ventrally to occupy the space left by the missing corpus callosum midline. The main difference between the two *Chd7* models is partial phenotypic penetrance in the *Whirligig* mouse, whilst the *Chd7^+/tm2a^* mutant exhibits systematic fully penetrant midline crossing failure. A previous study focusing on adult hippocampus neurogenesis using the same *Chd7^+/tm2a^* model made no mention of neuroanatomical defects in the hippocampus despite providing strong evidence that *Chd7* plays a role in hippocampal plasticity [15]. In line with this, we found no atrophy of the hippocampus in the *Chd7^+/tm2a^* model, while the *Whirligig* mouse shows midline crossing failure exhibited a worse phenotype with hippocampal distrophy (red dashed line in Figure 1E) and dilated dorsal third ventricle (red asterisk in Figure 1E), as previously reported [26]. Other neuroanatomical phenotypes were quantified only in the *Chd7^+/tm2a^* mutant since they were fully penetrant. The size of the mammillothalamic tracts increased by 100%, whilst the area of the internal capsule was decreased by 36%. Lesions of the mammillothalamic tract have been associated with impairments in temporal and contextual memory (reviewed in [27]), and numerous reports have linked motor functions such as swallowing (reviewed in [28]) or grip strength and hand coordination to the internal capsule. The later are exemplified by functional magnetic resonance imaging (fMRI) studies of stroke patients where motor defects associate with impaired internal capsule integrity, or in animal models with internal capsule lesions [29,30]. It is thus possible that the reduction in the internal capsule size may relate to the reduction in grip strength in *Chd7^+/tm2a^* mutants (Figure 3D).

Second, in addition to the neuroanatomical findings, our results provide a wider phenotypic spectrum which completes the abundance of reports focusing on the function of *Chd7*. *Chd7* mutants having shorter stature from birth fits with the reported ~65% of patients with CHARGE syndrome exhibiting this phenotype. Likewise, hyperactivity, such as that seen in mouse mutants has been reported in affected boys, and this behavior might be the cause of lower growth rates [31,32]. Unfortunately, hyperactivity is a major confounding factor for behavioral investigations. In the open-field test, the centre of the arena is anxiogenic and mice would tend to wander in the periphery, only occasionally exploring the centre. The *Chd7^+/tm2a^* mutants have reduced resting time, just as the *Chd7^+/Whi^* mutants, but the *Whirligig* mutants spent more time exploring the whole arena than their respective control animals. Hence, *Chd7^+/Whi^* either exhibit resistance to anxiety or this is the result of increased hyperactivity. Mice on a C3HeB/FeJ background (which is the case of the *Chd7^+/Whi^* mutant) are qualified as anxious and “jumpy” mice by the Jackson laboratory which, incidentally, makes them better models to evaluate anxiety resistance (but not hypersensitivity). Interestingly, the stress-induced hyperthermia paradigm is a model that studies the activation of the autonomic nervous system in response to stress by measuring body temperature. In this test, both *Chd7^+/tm2a^* and *Chd7^+/Whi^* mutants performed equally, which suggests that the mice are simply hyperactive and not resistant to anxiogenic environments.

From a metabolic perspective, *Chd7* mutants appear normal, although it should be emphasized that there is a bimodal distribution when we performed the glucose tolerance test in the *Chd7^+/Whi^* line, where mice present either improved or decreased glucose tolerance (Figure 4A,B). Such 50–50 patterns are common and typically occur in outbred populations [33], amongst many examples: Wistar rats have “low” and “high” responders in the test of cephalic insulin response to oral glucose [34], and Sprague Dawley rats present a dual sensitivity to diet-induced obesity [35]. These effects can easily go unnoticed as they rely on careful data description and having enough statistical power. Such dichotomic effects in an inbred context should not be discarded as they could indicate a drift. A very good example which received a lot of attention is how the C57BL/6J differs metabolically from the C57BL/6N strain [36].

Interestingly, basal insulin levels were reduced in both *Chd7^+/tm2a^* and *Chd7^+/Whi^* mutant lines. The only case of CHARGE syndrome with reported insulin levels showed hyperinsulinemia in infancy with severe hypoglycemia soon after birth [37]. However, *Chd7* expression is inversely proportional to *Sox4* expression [13,38] and in a knockout context, if *Chd7* is reduced, *Sox4* expression would be increased. Elevated *Sox4* expression has been shown to reduce insulin secretion by impaired fusion pore expansion in pancreatic β-cells [39], although we did not see an effect on glucose clearance. Current guidelines for CHARGE patient care do not require investigation of glucose/insulin parameters but in-depth exploration seems warranted. Blood chemistry highlighted stronger phenotypes in *Chd7^+/Whi^* compared to *Chd7^+/tm2a^* mutants, although trends generally followed similar patterns. Lipid parameters were all reduced. Currently, there are no guidelines for establishing lipid profiles in CHARGE patients. Lymphocyte immunophenotyping revealed subtle changes in Natural Killer (increased) and monocyte (decreased) cell counts. Immunodeficiency in CHARGE syndrome is rare, the severity of which depends largely on the degree of impairment in thymic development which impacts T-cell production [40]. In both mouse mutant lines, T-cell counts were normal, arguing these do not model CHARGE syndrome immunodeficiency well enough to warrant further investigations.

Third, our study makes it possible for mouse researchers interested in CHARGE syndrome to make informed choices on which mouse model they should use to study phenotypes of interest, and their underlying pathophysiological mechanisms. While both *Chd7^+/tm2a^* and *Chd7^+/Whi^* heterozygous LoF affect multiple organs (Figure 2), recapitulating core features of CHARGE syndrome, the inbred genetic background of *Chd7^Whi^* line associates with the most severe phenotypes, sometimes having reduced penetrance. Incomplete penetrance in *Chd7^+/Whi^* mutants has been well documented for other phenotypes (eye defects, genital anomalies, cardiovascular defects, cleft palate and choanal atresia), although it is noteworthy that not all phenotypes have the same degree of incomplete penetrance, which makes the *Chd7^+/Whi^* mutant a very interesting CHARGE syndrome model since individuals with CHARGE syndrome share the same principle of phenotypic variability. On the other hand, the mixed genetic background of *Chd7^+/tm2a^* mutant seems to offer resistance, with a reduced number of systems affected. It is significant that both models can therefore contribute differently to our understanding of the heterogeneity of *Chd7*-mediated diseases. Notably, the *Chd7^+/tm2a^* represents a reliable model to study corpus callosum dysgenesis. In CHARGE syndrome, corpus callosum anomalies are rarely reported, although there may still be a lack of guidelines for the neuroradiological evaluation of patients. For example, guidelines in CHARGE syndrome focusing on cranial imaging have been published [4], but callosal abnormalities are not listed as recommended neuroanatomical defects to search for. Anecdotally, the only MRI image shown of a CHARGE patient in this publication shows ventriculomegaly and thinning of the corpus callosum. A second extensive retrospective review of head and neck MRI findings in 10 CHARGE syndrome patients does not report callosal anomalies at all [41]. Taken together, our findings suggest that the genetic background of individuals diagnosed with CHARGE syndrome is the most likely source of phenotypic variation.

## 4. Materials and Methods

*Chd7^<tm2a(EUCOMM)Wtsi>^* mutant mice were generated by the European Conditional Mouse Mutagenesis Program (EUCOMM) and carried the knockout-first allele, a lacZ reporter-tagged insertion with conditional potential in C57BL6/NTac embryonic stem cells (Figure 1A), as described in White et al. [42]. The *Whirligig* mouse, *Chd7^Whi^*, was generated by a large-scale ENU-induced mutagenesis program [21].

*Chd7^Whi^* mice were genotyped by PCR (forward primer: 5′-ACTCAGGGAATACCAATTGGAG-3′; reverse primer: 5′-CAAAGAAAAGTTCCCAGCAAAC-3′) followed by an MfeI restriction digest. An MfeI recognition site was present in the wildtype sequence but not in the *Whirligig* mutant sequence. The forward primer had a single base pair mismatch which allows for the introduction of a recognition site for MfeI within the primer itself, providing an internal control for digestion. Digesting PCR products with MfeI generated a single band of 204 bp for wildtype (WT), two bands of 204 bp and 233 bp for *Chd7^+/Whi^*, and one band of 233 bp for *Chd7^Whi/Whi^* animals. For the *Chd7^tm2a^*, genotyping was performed using a forward (5′-TGCAGATGGGACGTTTTCAG-3′) and reverse primer (5′-TCGTGGTATCGTTATGCGCC-3′ for mutants and 5′-CTGCAAGAACACAGGGCAAG-3′ for WT). The quality control for the *Chd7^tm2a^* mutant was performed using the standard IMPC strategy, validated by loxP PCR confirmation, mutant specific PCR, homozygous loss of WT allele by qPCR and Neo count assay by qPCR to determine the copy number of the Neo Cassette. The results can be found here: https://www.mousephenotype.org/data/alleles/qc_data/mouse/MAWN/ (accessed on 26 September 2022).

Mice were fed on high-fat diet (21.4% crude fat content, Western RD, 829100, Special Diets Services, Witham, UK) from 4 weeks of age and weighed at regular intervals between 4 and 16 weeks of age. In total, 7 heterozygous male mice and 7 litter mate controls matched for sex and age were used in the case of the *Chd7^+/Whi^* cohort. For the *Chd7^+/tm2a^* cohort, 12 wild types were compared to 11 mutant male mice. Neuroanatomical studies were carried out with a smaller number of mice as recognized by our previous publication [17] where we showed that 3 mice per group is sufficient to evaluate neuroanatomical defects with an effect size of 10% or more at the 80% detection power threshold.

At week 9, anxiety and exploratory drive were assessed over 10 min using an open-field apparatus (PhenoMaster ActiMot; TSE Systems, Berlin, Germany). Mice were subsequently assessed for gross behavioural abnormalities using a modified SHIRPA screen. Qualitative scoring (normal versus abnormal) were performed for each animal and each of the following parameters: body position, palpebral closure, lacrimation, tremor, defecation, urination, transfer arousal, gait (including ataxia), pelvic elevation, tail elevation, locomotor activity, touch escape, startle response, positional passivity, trunk curl, limb grasping, evidence of biting, vocalization, pinna touch reflex, corneal touch reflex, contact righting reflex (Appendix A).

The same week, neuromuscular function and muscle strength were assessed with a Bioseb grip strength meter. Three trials each of fore paws and all paws were performed immediately following each other.

At week 10, mice were assessed for thermal pain perception using a hotplate (TSE systems). Each mouse was placed on the plate heated to 52 °C, and the latency to the first response, and response type, were recorded (Appendix A). Mice that did not respond after 30 s were removed.

At week 10, whole body morphology was examined using a standardised checklist (Appendix A). The coat and whiskers were examined using a standardised checklist at a separate time point (week 4). Hair follicle cycling was assessed at approximately 43 days of age.

At week 12, metabolic function was assessed using the LabMaster system (TSE-systems). Mice were housed individually in calorimetry cages for a period of approximately 21 h from approximately 2 pm. Cumulative food intake, activity (beam breaks), volume of oxygen consumed, and volume of carbon dioxide produced were all measured. From these data, the respiratory exchange ratio and energy expenditure were derived. VO_2_, VCO_2_, activity and energy expenditure parameters refer to an average during the total time period. Water bottles were weighed before and after the experiment to assess water consumption.

At week 13, mice were fasted overnight (maximum of 16 h) before a glucose tolerance test. A fasting blood sample was taken before a bolus of glucose administered by intra-peritoneal injection. Blood samples were tested for glucose concentration (Accu-Check Aviva, Roche) at 15, 30, 60 and 120 min following the glucose administration, and data presented as plasma glucose concentration.

At week 14, mice were anesthetized and imaged on a dual energy X-ray absorptiometry machine (Lunar PIXImus II). This generated an image of the entire mouse (minus the head) and provided bone mineral and body composition data.

Digital X-ray images were then acquired using a Faxitron system MX20 (Faxitron X-ray Corporation, Tucson, AZ, USA). Mice were anesthetized and up to five standard images were taken for each mouse. The images were annotated using a standardised protocol.

At week 15, basal core body temperature was measured rectally using a TH-5 thermometer with a RET-3 probe (Viking Medical, Clifton, NJ, USA). The mice were thereafter placed in a clean cage and a second reading was taken after 15–30 min. Mice were then assessed for gross morphological changes to the eye using a slit lamp (Zeiss SL130) and ophthalmoscope (Heine Omega 500). The eye was examined both undilated and dilated (tropicamide and/or neosynephrine). Images on the slit lamp were collected using a Leica DFC420. Images of the fundus were collected by using a topical endoscope (BERCI Tele-Otoscope with HOPKINS straight forward 0°, diameter 3mm, Halogen cold light fountain light source) and camera (Nikon D40x with Nikon AF 85mm F1.8D AF Nikkor lens).

At week 16, non-fasted mice were terminally anesthetized with ketamine/xylazine, and blood was collected into lithium/heparin coated tubes via the retro-orbital sinus. Plasma was analyzed for the following parameters on an Olympus AU400: electrolytes (sodium, potassium, chloride), non-fasted metabolic panel (glucose, fructosamine, triglycerides, cholesterol, high density lipoprotein, low density lipoprotein, non-esterified free fatty acids and glycerol), pancreatic enzyme (amylase), liver/muscle panel (alanine aminotransferase, alkaline phosphatase, creatine kinase, aspartate aminotransferase, total bilirubin), protein parameters (total protein, albumin), kidney panel (creatinine, urea), minerals and iron (calcium, magnesium, iron, phosphate). Plasma insulin concentration was measured by Mesoscale Discovery (MSD) array technology.

At week 16, blood was also collected into EDTA-coated tubes via the retro-orbital sinus. Whole blood was analyzed for the following parameters: white and red blood cell counts, mean corpuscular volume, hemoglobin, erythrocyte indices (haematocrit, mean corpuscular hemoglobin, mean corpuscular hemoglobin concentration, red blood cell distribution width), platelet counts and mean platelet volume. All blood samples were analysed with an automatic haematology analyzer using volume impedance principle (scilVet Animal Blood Counter, RAB 015 A Ind.E, 22.02.01).

Fluorescence activated cell sorting (FACS) analysis was performed on heparinized blood for the following parameters: percentages of total T cells, CD4+ and CD8+ T cells, NKT cells, NK cells, B cells, granulocytes and monocytes were expressed relative to the total CD45+ WBC population. Percentages of CD4+CD25+ regulatory T cells were presented relative to the total CD4+ T cell population. Percentages of mature IgD+ B cells were presented relative to the total B cell population. All samples were analyzed on a BD LSR II Flow Cytometer.

We bred a different cohort of mice, terminally anesthetized mice at 6 weeks of age for *Chd7^Whi^* and 8 weeks of age for *Chd7^tm2a^*, and had brain extracted and processed using a standardized histological pipeline at precise stereotaxic coordinates as described previously [25]. Briefly, brain samples were immersion-fixed in 10% formalin for 48 h, before paraffin embedding and sectioning at 5 μm thickness. A coronal section at Bregma −1.34 mm was double-stained (Luxol Fast Blue for myelin and Cresyl violet for neurons) and scanned at cell-level resolution using the Nanozoomer whole-slide scanner 2.0HT C9600 series (Hamamatsu Photonics, Shizuoka, Japan). In all, 41 parameters made of areas and lengths measurements pertaining to 14 independent brain regions were quantified using manual segmentation on ImageJ (https://imagej.net/software/fiji/, accessed on 26 September 2022) using scripted routines.

Statistical analyses were performed using the software R and Rstudio (https://www.rstudio.com, accessed on 26 September 2022). ANOVA (analysis of variance) with repeated measures were performed on parameters involving several measures across time. A two-tailed Student’s *t*-test assuming equal variance were carried out when normality tests reached significance and non-parametric tests used otherwise (Wilcoxon Mann–Whitney test).

## Figures and Tables

**Figure 1 ijms-23-11509-f001:**
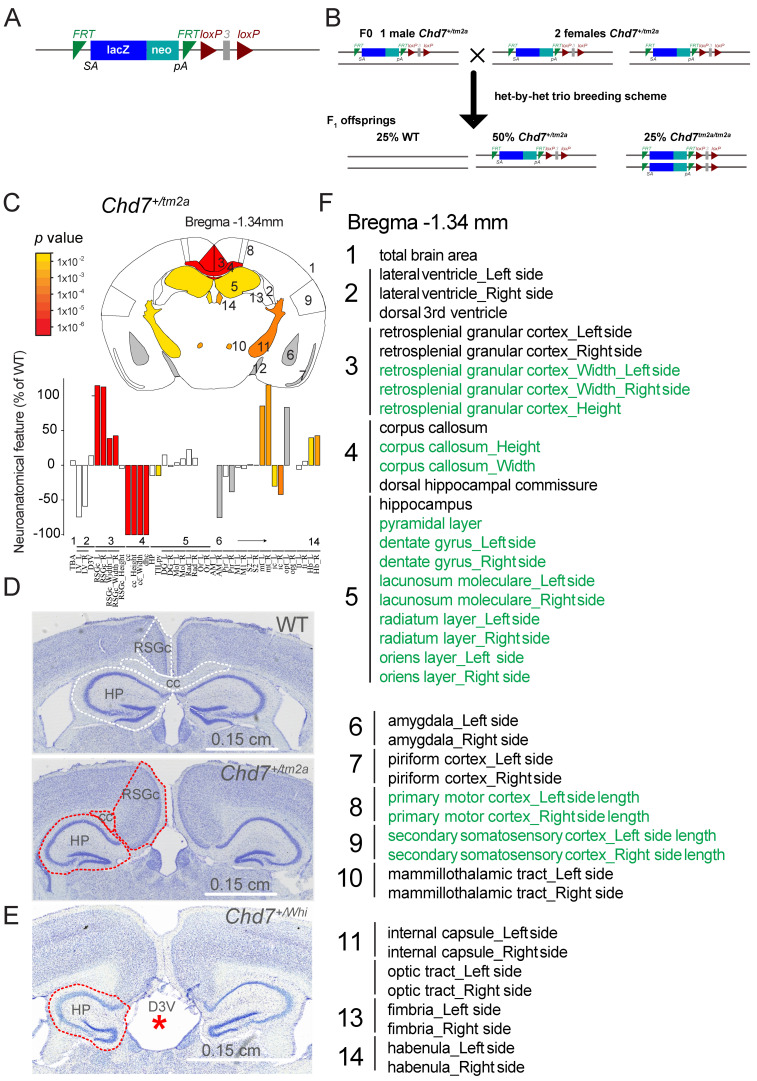
*Chd7* male mice exhibit major defects in midline crossing and fiber tracts. (**A**) Allelic construction of the *Chd7^tm2a^* mouse model. (**B**) Breeding strategy used in the *Chd7^tm2a^* mouse study. (**C**) The image shows a schematic representation of affected brain regions in *Chd7^+/tm2a^* male mice at 16 weeks of age plotted in the coronal plane (section at Bregma—1.34 mm) and colored according to *p* values on the side. Gray indicates parameters which could not be confidently tested (n too low to calculate the statistics). The name of each brain region numbered in (**C**) is given in panel (**F**). Histograms show neuroanatomical features as the percentage decrease (minus scale) or increase (plus scale) of the measured brain regions in *Chd7^+/tm2a^* mutant mice compared with matched controls. (**D**) Representative coronal brain sections of *Chd7^+/tm2a^* male mice and matched controls double-stained with Nissl and Luxol fast blue. Brain regions of interest are highlighted by white and red dashed lines in controls and mutants, respectively. (**E**) On a coronal plane at Bregma—1.34 mm, *Chd7^+/Whi^* mice show enlarged ventricles (indicated by a red asterisk), hippocampus dysgenesis (indicated by red dashed lines) and failure of midline crossing, albeit only in one sample, the other two appearing normal. (**F**) Details of brain regions assessed in order of appearance in panels (**C**) together with corresponding numbers. Green is for lengths and black is for areas. Retrosplenial granular cortex: RGCs, Hippocampus: HP, Dorsal third ventricle: D3V, Corpus callosum: CC.

**Figure 2 ijms-23-11509-f002:**
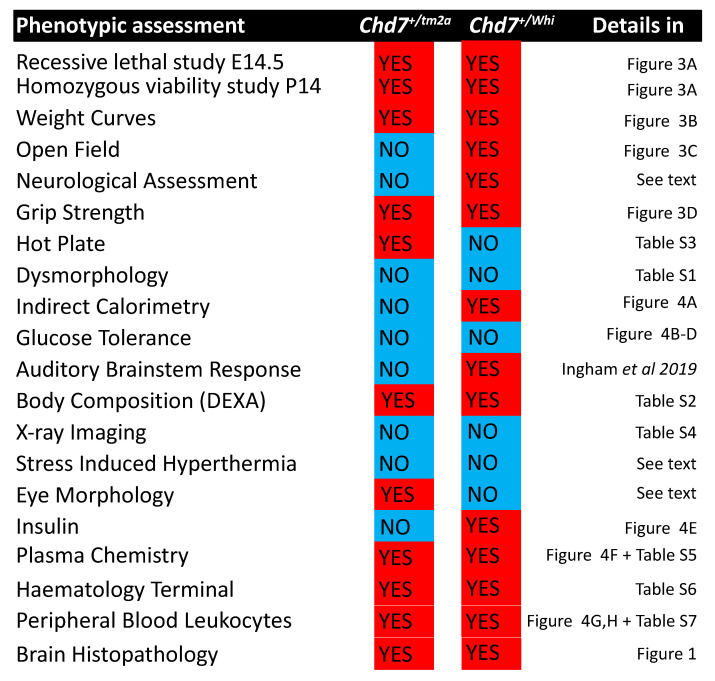
Result summary of phenotypic screen. Yes/red indicates significant threshold reached in the specified parameter. No/blue indicates normal phenotypes relative to respective line controls. The last column indicates where the detailed analyses are (figure, tables or references as auditory brainstem Response were previously published [19]).

**Figure 3 ijms-23-11509-f003:**
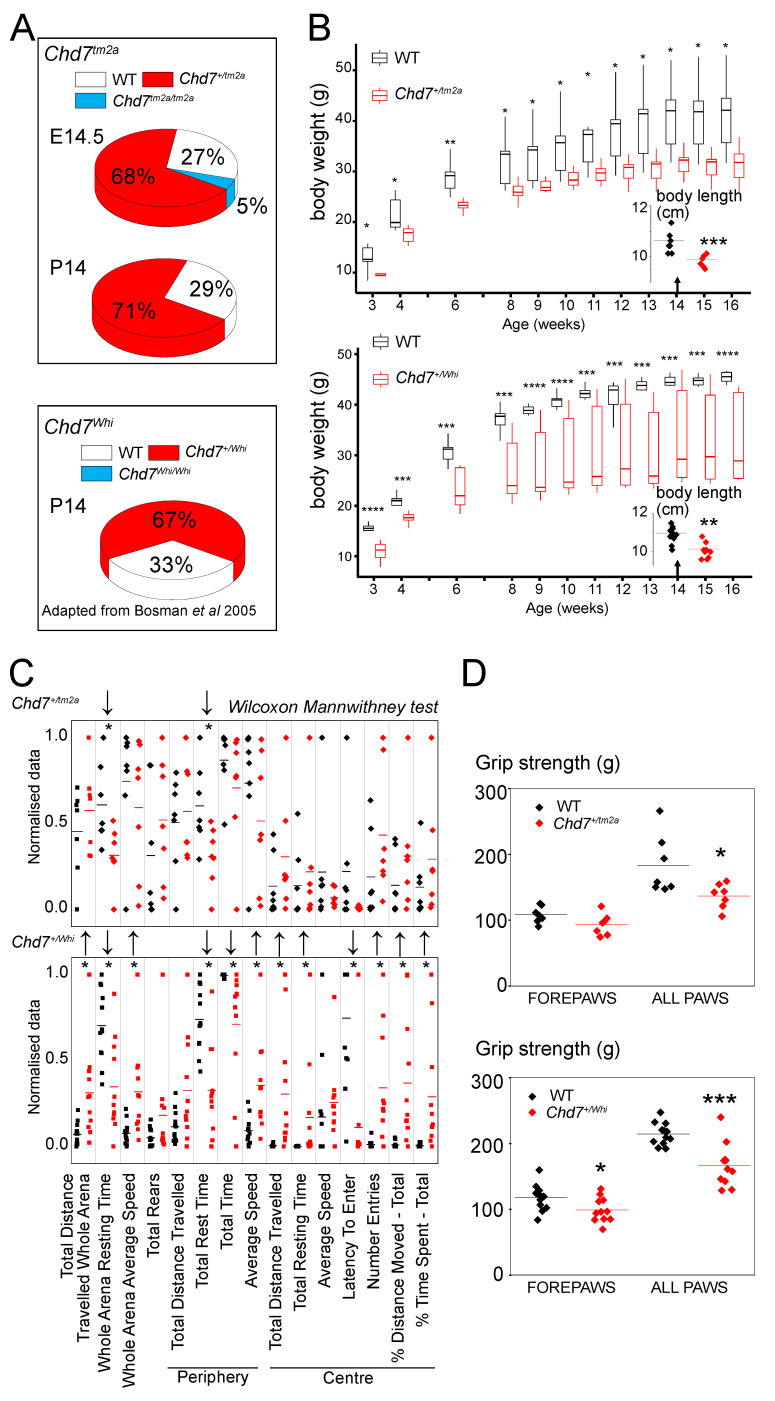
*Chd7^+/tm2a^* and *Chd7^+/Whi^* mice show similar defects in viability, growth, behavioral and grip strength, albeit to different levels. (**A**) Pie chart showing embryonic and after birth viability in *Chd7^+/tm2a^* (this study) and *Chd7^+/Whi^* (adapted from Bosman et al., 2005 [21]). (**B**) Body weight over time and body length at 14 weeks using box plots (SE) and whiskers for outlier boundaries. * *p* < 0.05, ** *p* < 0.005, *** *p* < 0.0005, **** *p* < 0.00005 using a one-way anova repeated measures. (**C**) Open field arena showing a pronounced increase in locomotor activity in *Chd7^+/Whi^* mice using a Wilcoxon mann-whitney test. (**D**) Mean and individual points showing severe (*Chd7^+/Whi^* mice) and mild (*Chd7^+/tm2a^*) decrease in grip strength.

**Figure 4 ijms-23-11509-f004:**
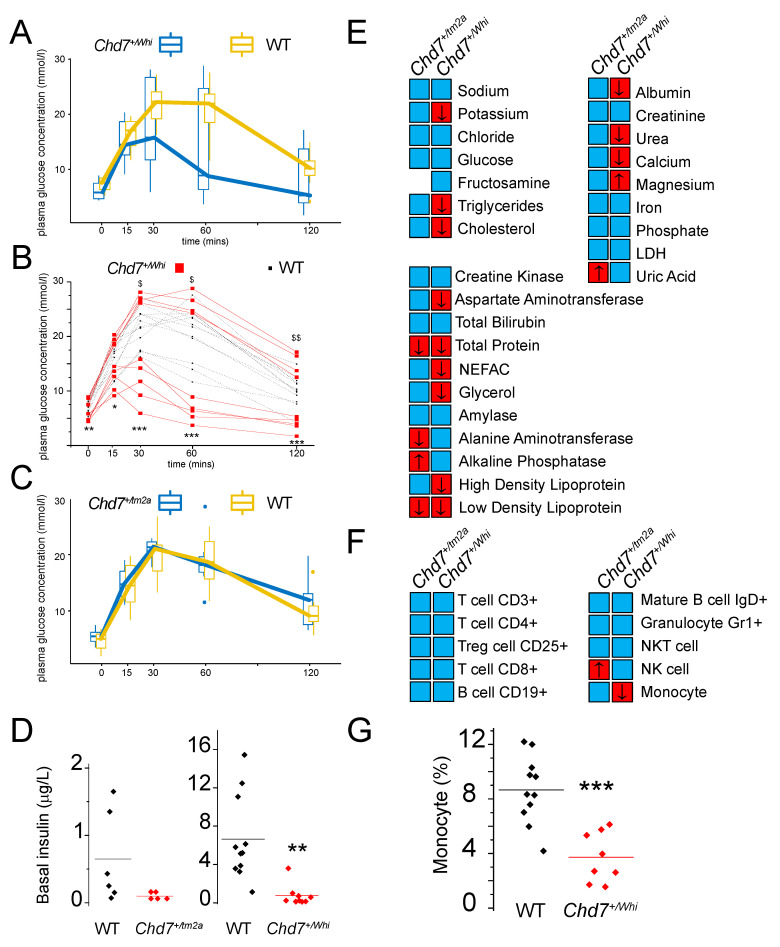
Metabolic and immunological parameters show dichotomic features between *Chd7^+/tm2a^* and *Chd7^+/Whi^* mutants. (**A**) Glucose tolerance expressed as mean ± sem (boxes) with outliers (whiskers) in *Chd7^+/Whi^* mice. (**B**) When the data are plotted as individuals, glucose tolerance in *Chd7^+/Whi^* mice show binomial distribution: two groups (high responders and low responders) can be identified. ^$^
*p* < 0.05 and ^$$^
*p* < 0.005 relative to controls in the high responder group. * *p* < 0.05, ** *p* < 0.005, *** *p* < 0.0005 relative to controls in the low responder group using anova repeated measures. (**C**) Box plots representing mean ± sem (boxes) with outliers (whiskers) in *Chd7^+/tm2a^* show normal glucose tolerance relative to their respective controls. (**D**) Basal insulin levels in *Chd7^+/tm2a^* and *Chd7^+/Whi^* mice. (**E**) Blood chemistry summary using a heatmap coded with blue (normal) and red (abnormal) and directionality (arrow going up for increased and down for decreased). Mean ± sem and t-test statistics are available in Appendix A. (**F**) Immunology screen summary illustrated using a heatmap and based on results available in Appendix A. Colour coded as in (**E**): blue (normal) and red (abnormal) and directionality (arrow going up for increased and down for decreased). (**G**) Monocyte mean and individual points for *Chd7^+/Whi^* mice. Statistics based on t-tests unless otherwise specified.

## Data Availability

We have provided all the data produced in this study within the article.

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
