# Peer review of "Characterization of Two Mouse Chd7 Heterozygous Loss-of-Function Models Shows Dysgenesis of the Corpus Callosum and Previously Unreported Features of CHARGE Syndrome"

_ijms, 2022, doi:10.3390/ijms231911509_

Round 1
Reviewer 1 Report
1. It is hard to understand how the authors generated the two different mouse lines. schematic figures describing mating strategies will help understanding this.
2. I am not sure why the authors used different fonts when indicating genotypes of mice. This should be consistent.
3. It is already well-known that hybrid strains are more resistant and tolerant to genetic or environmental insults. This limit the novelty of this paper.
Reviewer 2 Report
An interesting study entitled “Characterization of two mouse Chd7 heterozygous loss-of-function models show dysgenesis of the corpus callosum and previously unreported features of CHARGE syndrome” presented a good piece of work.
Major concern is the authors have never described about the confirmation on the used mouse models (Chd7 +/Whi and Chd7 +/tm2a ), engineered on different genetic background. They have to ensure about the genotype analysis of the mouse models used in the study.
The study is presented in a very organised way. I am wondering the Figure 3B bottom [(B) body weight over time and body length at 14 weeks. * p<0.05, ** p<0.005, *** p<0.0005, **** p<0.00005 using a one-way anova repeated measures.] significance on the 12th week, the box plot and its SD are clearly overlapping and the body weight of Chd7 +/Whi mice and WT clearly indicating that the weight of WT mice submerged in the weight of Chd7 +/Whi mice. Still authors have got *** p<0.0005. Looks not clear. Statement in the text “Both Chd7 +/Whi and the Chd7+/tm2a colonies displayed lower body weight from early life to adulthood (Figure 3B), and exhibited shorter length at 14 weeks of age (Figure 3B insert). Body composition analysis showed reduced lean mass by 16% in both mutant lines (p=0.04 and p=0.005 in Chd7+/Whi and Chd7+/tm2a mice, respectively) and reduced fat mass (-30 %, p=0.04) and fat percentage (-28%, p=0.03) specifically in Chd7+/Whi mice.” not describing the sudden fluctuation in the 12th week.
Authors should ensure that the actual p values are presented in the text as much as possible by replacing the generalised p values (*p<0.05, p<0.005, p<0.0005, p<0.00005), this will provide the exact significance for the readers.
Authors shall update the discussion with the lot of advancements in the year 2022. Very lest recent updates have been presented in the literature.
Round 2
Reviewer 1 Report
I strongly suggest a schematic figure describing mouse generation strategy, mating scheme even though the mice are generated and reported already. This makes readers much better understand the authors' paper and give positive impression. If this part is digested well, further contents are not well acceptable in my perspective.
Reviewer 2 Report
Response form the author is not visible in the response letter
>We thank Reviewer #2 for his/her concern. In fact, Reviewer #1 raised the same point which we addressed in a comprehensive way on page 2 above.
Round 3
Reviewer 1 Report
no comments requested
Author Response
We are grateful to the Reviewer for his/her time helping us to improve our manuscript.
Reviewer 2 Report
Revised MS can be accepted after moving the "Supplementary Figure 1: Allelic construction (A) and breeding strategy (B) of the Chd7tm2a mouse model." to the main text. Instead of Supplementary Figure 1 , it can be figure, that will give better understanding for the readers.
Author Response
We thank the Reviewer for his/her comment. We made changes to Figure 1 and added the allelic construction and the breeding scheme as new Panels A and B, respectively.